# LEARNING TO UNDO: TRANSFER REINFORCEMENT LEARNING UNDER STATE SPACE TRANSFORMATIONS

## ABSTRACT

Transfer learning in reinforcement learning (RL) has shown strong empirical success. In this work, we take a more principled perspective by studying when and how transferring knowledge between MDPs can be provably beneficial. Specifically, we consider the case where there exists an undo map between two MDPs (a source and a target), such that applying this map to the target's state space recovers the source exactly. We propose an algorithm that learns this map via regression on state feature statistics gathered from both MDPs, and then uses it to obtain the target policy in a zero-shot manner from the source policy. We theoretically justify the algorithm by analyzing the setting when the undo map is linear and the source is linearly-$Q^\star$ realizable, where our approach has strictly better sample complexity than learning from scratch. Empirically, we demonstrate that these benefits extend beyond this regime: on challenging continuous control tasks, our method achieves significantly better sample efficiency. Overall, our results highlight how shared structure between tasks can be leveraged to make learning more efficient.

## 1 INTRODUCTION

Reinforcement learning (RL) has seen rapid progress in recent years and has achieved strong performance in complex tasks, such as video games, locomotion, manipulation, and navigation (Mnih et al., 2015; Lee et al., 2019; Zhu et al., 2019). Despite this, learning high-quality policies from scratch typically requires millions of environment interactions, which limits the applicability of RL in real-world domains where data collection is costly or time-consuming (Dulac-Arnold et al., 2019).

Transfer learning aims to overcome this challenge by leveraging knowledge learned from source tasks to accelerate learning on related target tasks (Taylor & Stone, 2009). It has the potential to drastically improve sample efficiency by effectively reusing prior experience. While transfer learning has seen strong empirical success in RL (Zhu et al., 2020), we study the problem from a more principled lens, where we explicitly model the structured similarity between tasks and exploit it for transfer.

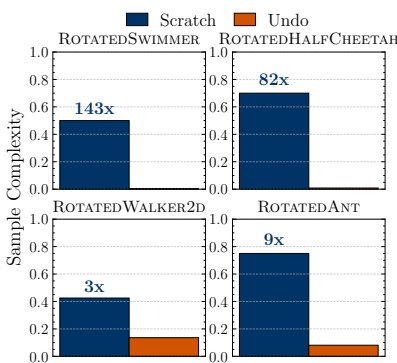

Figure 1: Sample complexity (in million) to reach 95% of optimal performance. Numbers on blue bars quantify how much worse learning from scratch is.

An interesting and practically relevant setting is that of state space transformations: the source and target tasks differ due to transformations in the state space. For instance, real-world transformations such as color transformation from RGB to grayscale, change in frame of reference, and sensor fusion – all are state space transformations (see Fig. 2). Such transformations provide a rich and practical setting to design principled transfer learning methods that explicitly model them.

In this work, we propose a novel approach to address this setting by learning an *undo map* that transforms the state space in the target back to the source. Instead of learning a policy from scratch, our method recovers a function that *undoes* the transformation applied to the target states. This allows us to reuse the source policies with minimal additional samples from the target task.

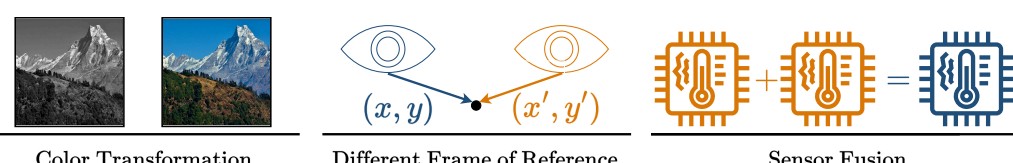

Figure 2: State space transformations are grounded in real-world applications.

We propose an algorithm that exploits the invariance of the action space to learn this undo map via regression on state feature statistics gathered from both MDPs. It subsequently uses the learned undo map to obtain the target policy in a zero-shot manner by composing the source policy with the undo map. Our algorithmic design is grounded in the setting where the undo map is linear and the source is linearly-$Q^\star$ realizable. In this case, we theoretically show that our approach has strictly better sample complexity than learning from scratch. Building on these insights, we extend our approach to handle non-linear state transformations without requiring additional assumptions on the MDPs.

To evaluate our method, we use challenging continuous control tasks, and construct target tasks by applying transformations such as changing the frame of reference, changing the coordinate system, settings inspired by sensor fusion, as well as moving from sensor observations in the source to pixel observations in the target. Our results show that learning the undo map consistently outperforms learning the target policy from scratch in terms of sample efficiency, where it often achieves near-optimal performance with an order of magnitude fewer samples (see Fig. 1). These results highlight the practical utility of our approach for transfer learning in RL under state-space transformations.

To summarize, our work makes the following key contributions:

I. We propose a novel transfer learning method, LUMOS (**L**earning to **U**ndo by **M**atching **O**bservation **S**tatistics), that performs regression on state feature statistics to *undo* state-space transformations in the target MDP (Sections 3 and 4).

II. We ground our algorithmic design in the setting where the undo map is linear and the source MDP is linearly-$Q^\star$ realizable. Theoretically, we prove that our approach, LUMOS-LIN, achieves strictly better sample complexity than learning from scratch (Section 3).

III. Building on these insights, we propose an extension, LUMOS, that works for non-linear state-space transformations without additional assumptions on the MDPs (Section 4).

IV. We demonstrate the strong empirical performance of our method by conducting experiments on challenging continuous control tasks involving state-space transformations (Section 5).

## 2 PROBLEM SETUP

**Tasks.** We model each task as a finite-horizon episodic Markov Decision Process (MDP), defined by the 7-tuple $(\mathcal{S}, \mathcal{A}, \mathbb{P}, r, H, d_0, \phi)$, where $\mathcal{S}$ is the state space, $\mathcal{A}$ is the action space, $\mathbb{P} : \mathcal{S} \times \mathcal{A} \to \mathcal{P}(\mathcal{S})$ defines the transition dynamics, $r : \mathcal{S} \times \mathcal{A} \to \mathbb{R}$ is the reward function, $H$ is the fixed horizon length, $d_0 \in \mathcal{P}(\mathcal{S})$ is the initial state distribution, and $\phi : \mathcal{S} \to \mathbb{R}^d$ is a fixed feature map. We assume that $d \leq H$. The agent interacts with the task using a non-stationary policy $\pi_h : \mathbb{R}^d \to \mathcal{P}(\mathcal{A})$, and for $0 \leq h < H$, chooses an action $a_h \sim \pi_h(\cdot \mid \phi(s_h))$. The value of the policy $\pi$ is defined as the expected return $V^\pi \triangleq \mathbb{E}_{s_0 \sim d_0, \pi, \mathbb{P}} \left[ \sum_{h=0}^{H-1} r(s_h, a_h) \right]$, and the agent's goal is to learn a policy $\pi^\star$ that maximizes this quantity, i.e., $V^* = \max_\pi V^\pi$. The state-action value function $Q^\pi(s, a)$ denotes the expected return when taking action $a$ in state $s$, and following policy $\pi$ thereafter.

**Transfer Learning.** The transfer learning setting follows the protocol in which the agent first interacts with a source task $\mathcal{M}_S = (\mathcal{S}^S, \mathcal{A}^S, \mathbb{P}^S, r^S, H, d_0^S, \phi_S)$ to learn a policy $\pi_S$, and then leverages the learned policy to speed up learning in the target task $\mathcal{M}_T = (\mathcal{S}^T, \mathcal{A}^T, \mathbb{P}^T, r^T, H, d_0^T, \phi_T)$. We assume that the agent has complete access to the source MDP $\mathcal{M}_S$, either by knowing its dynamics and reward function exactly or by being allowed to interact with it arbitrarily without incurring any

sample complexity cost. In contrast, interactions with the target MDP $\mathcal{M}_T$ are limited and costly. $\mathcal{M}_S$ and $\mathcal{M}_T$ have some shared structure, which we describe below.

**Shared Structure and Undo Map.** We assume that both tasks are exactly the same except that the agent observes states through task-specific feature maps: $\phi_S : \mathcal{S} \to \mathbb{R}^{d_S}$ in the source and $\phi_T : \mathcal{S} \to \mathbb{R}^{d_T}$ in the target. These feature maps are related by an unknown transformation (the *undo* map) $U_\star : \mathbb{R}^{d_T} \to \mathbb{R}^{d_S}$, such that for all $s \in \mathcal{S}$, $\phi_S(s) = U_\star\big(\phi_T(s)\big)$. This relationship induces a correspondence between policies: for any policy $\pi_S : \mathbb{R}^d \to \mathcal{P}(\mathcal{A})$ defined in the source feature space, we define the corresponding policy in the target task as $\pi_T\big(\phi_T(s)\big) = \pi_S\Big(U_\star\big(\phi_T(s)\big)\Big)$. This allows policies to generalize across tasks despite differing state representations.

**Objective.** Given complete access to the source MDP $\mathcal{M}_S$, our objective is to learn a near-optimal policy for the target MDP $\mathcal{M}_T$ using as few samples from the target as possible.

## 3 LUMOS-LIN: LEARNING THE UNDO MAP IN LINEAR SETTINGS

In this section, we focus on linear state-space transformations, and show how the undo map can be estimated from samples by leveraging the invariance of the action space in the source and target MDPs (Section 3.1). The resulting learning algorithm, LUMOS-LIN, computes state feature statistics in the source and target that are related via the undo map. This algorithm, illustrated in Fig. 3, enjoys strictly better sample complexity than learning from scratch (Section 3.2) and enables zero-shot transfer of the target policy by composing the source policy with the learned undo map.

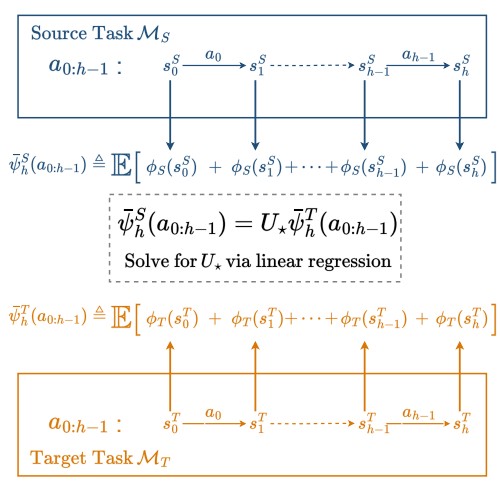

Figure 3: Overview of LUMOS-LIN. We recover the map to *undo* the state space transformation by linear regression on state feature statistics.

### 3.1 LEARNING TO UNDO VIA REGRESSION

To estimate the undo map in linear settings, we compare statistics of state features under an *open-loop* sequence of actions in the source and target MDPs. While the observations differ, the underlying dynamics remain consistent.

**Definition 1** (Expected State Feature Sum).
*Given an action sequence $a_{0:h-1}$, define $\psi_h(a_{0:h-1})$ as the $h$-step truncated sum of state features computed from a sample trajectory:*

$$\psi_h(a_{0:h-1}) \triangleq \sum_{t=0}^{h-1} \phi(s_t), \tag{1}$$

*where $(s_0, s_1, \ldots, s_{h-1})$ is the state sequence obtained by executing actions $a_0, \ldots, a_{h-1}$. The expected state feature sum for $h$ steps, denoted by $\bar{\psi}_h(a_{0:h-1})$, is defined as:*

$$\bar{\psi}_h(a_{0:h-1}) \triangleq \mathbb{E}\left[\psi_h(a_{0:h-1})\right], \tag{2}$$

*where the expectation is over the dynamics of the MDP.*

Note that the expected state feature sum is defined with respect to an *open-loop* sequence of actions. Consequently, since the source and target MDPs differ only through a linear transformation of the state space, the expected state feature sums in the source and the target (denoted by $\bar{\psi}_h^S(a_{0:h-1})$ and $\bar{\psi}_h^T(a_{0:h-1})$, respectively) are related via the undo map $U_\star$ as follows:

$$\bar{\psi}_h^S(a_{0:h-1}) = U_\star \bar{\psi}_h^T(a_{0:h-1}), \ \forall h \in [H]. \tag{3}$$

---

**Algorithm 1** Learning to Undo with LUMOS-LIN

---

**Require:** Source MDP $\mathcal{M}_S$, target MDP $\mathcal{M}_T$, action sequence $a_{0:H-1}$, number of trajectories $n$
1: Exactly compute the expected feature sum $\bar{\psi}_h^S(a_{0:h-1})$ in $\mathcal{M}_S$, $\forall h \in [H]$.
2: Roll out $n$ trajectories in $\mathcal{M}_T$ using action sequence $a_{0:H-1}$.
3: **for** $h \in [H]$ **do**
4:     For each trajectory $i$, compute feature sum $\psi_h^{T,(i)}(a_{0:h-1})$.
5:     Compute empirical expectation: $\widehat{\psi}_h^T(a_{0:h-1}) = \frac{1}{n}\sum_{i=1}^n \psi_h^{T,(i)}(a_{0:h-1})$.
6: **end for**
7: Solve for $\widehat{U} \in \mathbb{R}^{d_S \times d_T}$ (which estimates $U_\star$) via least squares:

$$\widehat{U} = \arg\min_{U \in \mathbb{R}^{d_S \times d_T}} \sum_{h=1}^H \left\| \bar{\psi}_h^S(a_{0:h-1}) - U\widehat{\psi}_h^T(a_{0:h-1}) \right\|_2^2.$$

8: **return** $\widehat{U}$

---

As samples from the source MDP $\mathcal{M}_S$ are free, we can compute $\bar{\psi}_h^S(a_{0:h-1})$ exactly, while $\bar{\psi}_h^T(a_{0:h-1})$ can be empirically estimated using samples from the target MDP $\mathcal{M}_T$.

**Learning Algorithm.** The relations between expected state feature sums in the source and the target are equivalent to a system of linear equations (one for each time horizon $h \in [H]$), which we use to estimate $U_\star$ via least squares. Since $U_\star \in \mathbb{R}^{d_S \times d_T}$, estimating it amounts to solving $d_S$ separate linear regression problems, each with $d_T$ unknowns. Therefore, we need at least $d_T$ linearly independent equations to ensure identifiability. Since $d_T < H$ (by assumption), we use the expected feature sum relation for each $h \in [H]$ to construct $H$ such equations. Specifically, let $\widehat{\psi}_h^T(a_{0:h-1})$ denote the empirical estimate of the expected target feature sum obtained from $n$ trajectories. Then the undo map is estimated by solving:

$$\widehat{U} = \arg\min_{U \in \mathbb{R}^{d_S \times d_T}} \sum_{h=1}^H \left\| \bar{\psi}_h^S(a_{0:h-1}) - U\widehat{\psi}_h^T(a_{0:h-1}) \right\|_2^2. \tag{4}$$

**Lifting Source Policy to Target.** The estimate $\widehat{U}$ allows us to *lift* the optimal policy for the source task $\pi_S^\star$ to an optimal policy for the target task: $\pi_T^\star(\cdot) = \pi_S^\star\left(\widehat{U}(\cdot)\right)$.

The pseudocode for the proposed algorithm, LUMOS-LIN, is given in Algorithm 1.

## 3.2 THEORETICAL ANALYSIS

We first list the assumptions used in our analysis. The source and target MDPs have feature maps $\phi_S : \mathcal{S}^S \times \mathcal{A} \to \mathbb{R}^{d_S}$ and $\phi_T : \mathcal{S}^T \times \mathcal{A} \to \mathbb{R}^{d_T}$ that represent state–action pairs as $d_S$ and $d_T$-dimensional vectors. Let $\Phi_S \in \mathbb{R}^{|\mathcal{S}^S \times \mathcal{A}| \times d_S}$ and $\Phi_T \in \mathbb{R}^{|\mathcal{S}^T \times \mathcal{A}| \times d_T}$ denote the corresponding feature matrices, with rows $\phi_S(s,a)$ and $\phi_T(s,a)$. We assume that the feature vectors in both MDPs satisfy $\|\phi(s,a)\|_2 \le 1$, and that rewards are bounded within the range $[-1,1]$. We further assume that the source is linearly-$Q^\star$ realizable.

**Assumption 1** (Linear-$Q^\star$ realizability in $\mathcal{M}_S$ (Weisz et al., 2021)). *There exists a vector $\theta_S^\star \in \mathbb{R}^{d_S}$ such that for every state–action pair $(s,a) \in \mathcal{S}^S \times \mathcal{A}$ in the source MDP $\mathcal{M}_S$,*

$$Q_S^\star(s,a) = \langle \phi_S(s,a), \theta_S^\star \rangle.$$

In this setup, given an action sequence $a_{0:h-1}$, we define $\psi_h(a_{0:h-1})$ to be the average of the state-action features over the first $h$ steps of a sample trajectory:

$$\psi_h(a_{0:h-1}) \triangleq \frac{1}{h}\sum_{t=0}^{h-1} \phi(s_t, a_t). \tag{5}$$

The source and the target MDPs are exactly the same except for the state-action features, which are related via a linear undo map $U_\star$: $\Phi_S = \Phi_T U_\star^T$. We assume that the $\ell_2$ norm of each row of $U_\star$ is

upper bounded by a constant. The learner has access to the parameter vector $\theta_S^\star$ for the state-action values of the optimal source policy, and an oracle $\mathcal{O}$ for the state-action feature sum in the source.

**Assumption 2** (Oracle for state-action feature sum in $\mathcal{M}_S$). *We denote by $\mathcal{O}$ an oracle that when queried with an action sequence $a_{0:h-1}$ and time $h \in [H]$, outputs $\bar\psi_h^S(a_{0:h-1}) \triangleq \mathbb{E}\left[\psi_h^S(a_{0:h-1})\right]$.*

**Learning Algorithm.** Our learning algorithm first queries $\mathcal{O}$ to obtain $\bar\psi_h^S(a_{0:h-1})$ in $\mathcal{M}_S$, for all $h \in [H]$. Next, it rolls out $n$ trajectories in the target using the open-loop action sequence $a_{0:H-1}$. It then views the truncated sum of state-action features $\psi_h^{T,(i)}(a_{0:h-1})$ (computed on the $i$-th sample trajectory) as noisy features for $\bar\psi_h^S(a_{0:h-1})$. Subsequently, it estimates $U_\star$ with $\widehat{U}$ via least squares.

**Policy Transfer.** From Eq. 3, the relation $Q_T^\star = \Phi_T U_\star^T \theta_S^\star$ holds. Thus, a natural way to evaluate the output $\widehat{U}$ of LUMOS-LIN, is to bound the performance difference with respect to the optimal policy in the target, if we act greedily with respect to $\Phi_T \widehat{U}^T \theta_S^\star$. Let $\hat\pi$ denote this policy. We show that our algorithm has the following performance difference guarantee; the proof is provided in Appendix B.

**Theorem 1** (Performance Difference Bound for LUMOS-LIN). *Define the covariance matrix*

$$\Sigma \triangleq \frac{1}{H} \sum_{h=1}^{H} \psi_h^T(a_{0:h-1}) \psi_h^T(a_{0:h-1})^T,$$

*and let $\lambda_{\min}$ denote $\lambda_{\min}(\Sigma)$.*

*For all $n \geq \frac{6}{H\lambda_{\min}}(\log \frac{3 d_T d_S}{\delta})$, with probability at least $1 - \delta$, the performance difference satisfies*

$$\|V_T^\star - V_T^{\hat\pi}\|_\infty \leq 2\sqrt{\frac{H}{n\lambda_{\min}}} \sqrt{d_T + 2\log\left(\frac{3 d_S}{\delta}\right) + 2\sqrt{d_T \log\left(\frac{3 d_S}{\delta}\right)}} + o\left(\sqrt{\frac{H}{n\lambda_{\min}}}\right).$$

The theorem shows that if the *open-loop* action sequence $a_{0:H-1}$ is *good*, i.e., $\lambda_{\min}(\Sigma)$ is sufficiently large (on the order of $1/d_T$), then LUMOS-LIN achieves a sample complexity upper bound that scales as $\widetilde{O}(d_T\sqrt{H})$. This improves upon the lower bound for linearly-$Q^\star$ realizable MDPs, which scales exponentially with the feature dimension $d_T$ or the horizon $H$ (Weisz et al., 2021).

**Remark on Linear MDPs.** A special case of MDPs with linear-$Q^\star$ realizability is linear MDPs (Jin et al., 2020). In this setting, the minimax lower bound scales as $\widetilde{O}(d_T H^{3/2})$ (He et al., 2023), whereas our transfer learning method enjoys a $\widetilde{O}(d_T\sqrt{H})$ dependence.

## 4 LUMOS: LEARNING THE UNDO MAP IN GENERAL SETTINGS

In this section, we relax the assumption that the undo map is linear and present a general algorithm for non-linear transformations of the state-space.

Algorithm 1 does not directly apply in this setting, because the key relation in Eq. 3 no longer holds once $U_\star$ is non-linear. However, we can reinterpret what the algorithm was essentially doing: matching the expected state feature sums in the source MDP $\mathcal{M}_S$ with those computed from the *undone* trajectories in the target MDP $\mathcal{M}_T$.

With this lens, we introduce the notion of an expected *undone* feature sum, defined with respect to an undo map $U$, that looks at *undone* trajectories.

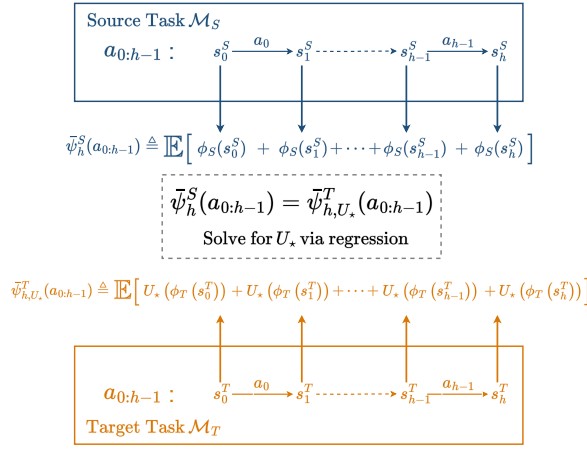

Figure 4: Overview of LUMOS. We match state feature statistics between source and *undone* target trajectories.

---

**Algorithm 2** Learning to Undo with LUMOS

---

**Require:** Source MDP $\mathcal{M}_S$, target MDP $\mathcal{M}_T$, number of trajectories $n$, number of iterations $N$, truncated horizon $H^\dagger$
1: Initialize $\phi$.
2: **for** $i \in [N]$ **do**
3:     Sample an action sequence $a_{0:H^\dagger-1}$ uniformly at random.
4:     Exactly compute the expected feature sum $\bar{\psi}_h^S(a_{0:h-1})$ in $\mathcal{M}_S$, $\forall h \in [H^\dagger]$.
5:     Roll out $n$ trajectories in $\mathcal{M}_T$ using action sequence $a_{0:H^\dagger-1}$.
6:     **for** $h \in [H^\dagger]$ **do**
7:         For each trajectory $j$, compute undone feature sum $\psi_{h,U_\phi}^{T,(j)}(a_{0:h-1})$.
8:         Compute empirical expectation: $\widehat{\psi}_{h,U_\phi}^T(a_{0:h-1}) = \frac{1}{n}\sum_{j=1}^n \psi_{h,U_\phi}^{T,(i)}(a_{0:h-1})$.
9:     **end for**
10:     Update $\phi$ to minimize $\sum_{h=1}^{H^\dagger}\|\bar{\psi}_h^S(a_{0:h-1}) - \widehat{\psi}_{h,U_\phi}^T(a_{0:h-1})\|_2^2$.
11: **end for**
12: **return** $\phi$

---

**Definition 2** (Expected Undone State Feature Sum). *Given an action sequence $a_{0:h-1}$ and an arbitrary undo map $U$, define $\psi_{h,U}(a_{0:h-1})$ as the $h$-step truncated sum of undone state features computed from a sample trajectory:*

$$\psi_{h,U}(a_{0:h-1}) \triangleq \sum_{t=0}^{h-1} U\left(\phi\left(s_t\right)\right), \tag{6}$$

*where $(s_0, s_1, \ldots, s_{h-1})$ is the state sequence obtained by executing actions $a_0, \ldots, a_{h-1}$. The expected undone state feature sum for $h$ steps, denoted by $\bar{\psi}_{h,U}(a_{0:h-1})$, is defined as:*

$$\bar{\psi}_{h,U}(a_{0:h-1}) \triangleq \mathbb{E}\left[\psi_{h,U}(a_{0:h-1})\right], \tag{7}$$

*where the expectation is over the dynamics of the MDP.*

By the definition of $U_\star$, we have

$$\bar{\psi}_h^S(a_{0:h-1}) = \bar{\psi}_{h,U_\star}^T(a_{0:h-1}), \forall h \in [H]. \tag{8}$$

Similar to the previous setting, we can compute $\bar{\psi}_h^S(a_{0:h-1})$ exactly, while $\bar{\psi}_{h,U}^T(a_{0:h-1})$ can be empirically estimated using samples from the target MDP $\mathcal{M}_T$ for any undo map $U$.

**Learning Algorithm.** In the general setting, we parameterize the undo map as $U_\phi$ and optimize its parameters $\phi$ to match the expected state feature sums of the source and the *undone* target trajectories. More concretely, we minimize the objective

$$\mathcal{J}(\phi) = \mathbb{E}_{a_{0:H^\dagger-1}}\left[\sum_{h=0}^{H^\dagger-1}\left\|\bar{\psi}_h^S(a_{0:h-1}) - \bar{\psi}_{h,U_\phi}^T(a_{0:h-1})\right\|_2^2\right] \tag{9}$$

with respect to $\phi$, where the expectation is over action sequences sampled uniformly at random.

Since the undo map is no longer linear, the $d_S \cdot d_T$ equations from a single action sequence are insufficient to estimate $U_\star$. Therefore, our algorithm samples multiple action sequences and optimizes the empirical expectation over these sequences. Moreover, we use a truncated horizon $H^\dagger < H$ to allow more action sequences to be sampled for training under a given budget of samples; we justify this choice in Section 5.

The pseudocode for our algorithm, LUMOS, is given in Algorithm 2. In summary, we iteratively sample an action sequence, compute the expected feature sums in the source, estimate the *undone* feature sums in the target via $n$ trajectories, and updates $\phi$ to minimize the empirical squared error.

Zero-shot transfer works as usual; compose the source policy with the learned undo map.

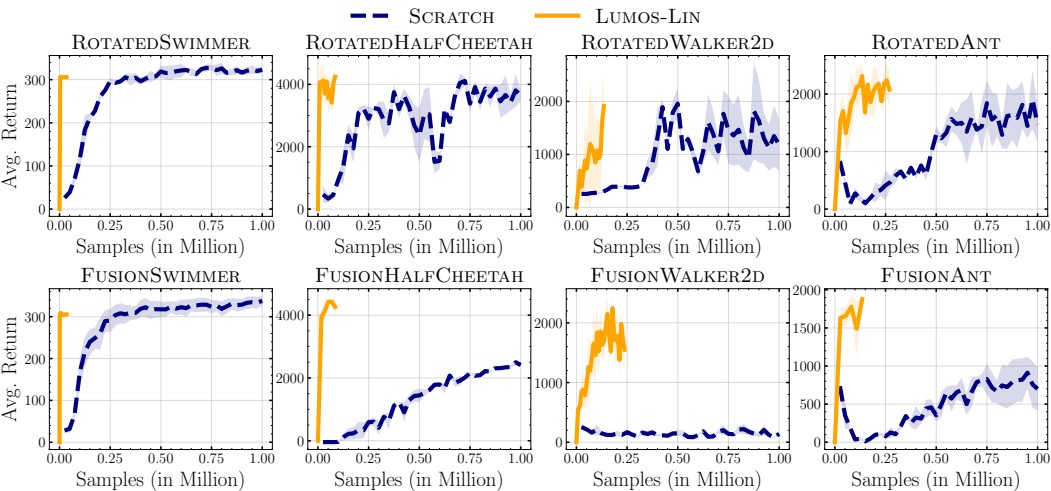

Figure 5: Comparison of avg. reward with varying budget of target samples. Transfer learning via LUMOS-LIN consistently achieves significantly better sample efficiency than learning from scratch.

## 5 EXPERIMENTAL EVALUATION

In this section, we empirically evaluate our transfer learning method to address the following research questions: (i) Does learning the undo map $U_\star$ outperform learning the target policy from scratch? (ii) How should the *open-loop* action sequence $a_{0:H-1}$ be chosen?

We begin by explaining the rationale for task selection, describing both the source and target tasks. Next, we provide an overview of the training process for the undo map, followed by the results.

### 5.1 ENVIRONMENTS

We evaluate our method on a suite of continuous control environments that are challenging to learn from scratch. As source tasks, we use standard MuJoCo environments: SWIMMER, HALFCHEE-TAH, WALKER2D, ANT, and CARTPOLE. We then construct corresponding target tasks by applying structured transformations in the state space. Detailed descriptions are provided in Appendix C.

**Changed frame of reference.** In setting, we create rotated variants of source tasks, where the global coordinate frame is rotated by $45°$. This results in ROTATEDSWIMMER, ROTATEDHALFCHEETAH, ROTATEDWALKER2D, and ROTATEDANT environments.

**Sensor fusion.** In this setting, multiple sensor modalities are present. Instead of observing a $d_S$-dimensional observation vector, the agent receives a $10 \times d_S$-dimensional observation. Concretely, for each source observation coordinate $i$, the target provides 10 measurements corresponding to different sensors, which can be linearly combined to recover the original value. This results in FU-SIONSWIMMER, FUSIONHALFCHEETAH, FUSIONWALKER2D, and FUSIONANT environments.

**Changed coordinate system.** In this setting, we construct polar-coordinate variants of HALFCHEE-TAH and ANT, where sensor readings are expressed in polar rather than Cartesian coordinates. This results in POLARHALFCHEETAH, and POLARANT environments.

**Pixel observations.** In this setting, we consider CARTPOLE with pixel observations as the target task, while the source task is sensor-based. This results in the PIXELCARTPOLE environment.

### 5.2 TRAINING PROCEDURE

We use the `rl-baselines3-zoo` framework (Raffin, 2020) to train both the source policies and the target policies used in the learning-from-scratch baseline, with PPO (Schulman et al., 2017) as the learning algorithm. For each environment, we adopt the tuned hyperparameters provided in the framework, and train the policies for 1 million steps each. To estimate the expected state feature statistics, we use a large number of samples for the source MDP to obtain accurate estimates, while

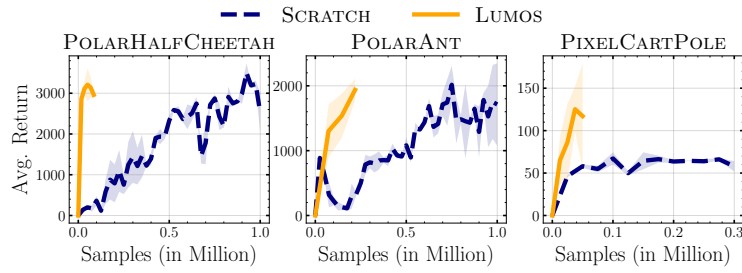

Figure 6: Comparison of avg. reward with varying budget of target samples. Transfer learning via LUMOS consistently achieves significantly better sample efficiency than learning from scratch.

using fewer samples for the target MDP. For LUMOS-LIN, the action sequence $a_{0:H-1}$ is derived from the source policy $\pi_S$: we roll out $\pi_S$ in the source MDP $\mathcal{M}_S$, record the sequence of actions taken, and reuse this sequence in both MDPs. For LUMOS, by contrast, we employ multiple short action sequences sampled uniformly at random. In this setting, the undo map is parameterized by a neural network. Additional implementation details are provided in Appendix D.

## 5.3 RESULTS

In this section, we first compare our methods, LUMOS-LIN and LUMOS, against learning the target policy from scratch (listed as SCRATCH). Next, we evaluate the criticality of the choice of the action sequence.

**Results with LUMOS-LIN.** While LUMOS-LIN comes with guarantees under linear undo maps and linearly-$Q^\star$ realizable MDPs, we evaluate it in settings where this assumption no longer holds. The ROTATED and FUSION environments fit this regime. Fig. 5 reports the average reward, averaged over three seeds, under varying target sample budgets. We observe

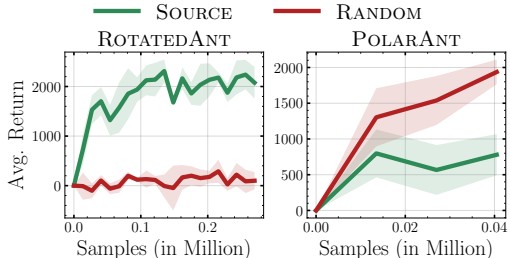

Figure 7: Sensitivity to action sequence. The choice of *open-loop* action sequence critically affects the performance of our method.

consistent and substantial gains over training from scratch. Notably, in ROTATEDSWIMMER, LUMOS-LIN is $143\times$ more sample-efficient. The FUSION tasks are particularly difficult for learning from scratch, likely due to the increased dimensionality of the state space and the fact that information is only meaningful when aggregated linearly across sensors. In contrast, LUMOS-LIN exploits this structure, and achieves near-optimal performance with very few target samples.

**Results with LUMOS.** We now consider the general non-linear transformation setting, without any realizability assumptions on the target MDP. Fig. 6 reports results on POLARHALFCHEETAH, POLARANT, and PIXELCARTPOLE. These environments are challenging: in practice, when angles appear in the observation space, they are often represented as (sin, cos) pairs to aid learning, whereas here we provide only raw polar coordinates. Despite this, LUMOS effectively recovers the ground-truth undo map, and leads to substantial sample efficiency improvements. Similarly, in PIXELCARTPOLE, LUMOS achieves higher average reward within a fraction of the target sample budget compared to learning from scratch.

Overall, these results demonstrate that LUMOS leverages the shared structure between source and target tasks, and consequently, leads to drastic improvements in sample efficiency across both linear and non-linear transformations.

**Choice of Action Sequence.** LUMOS-LIN requires a single *good* action sequence to estimate the undo map. We compare a source-anchored sequence and a randomly sampled sequence on RO-TATEDANT in Fig. 7 (left), and find that the source-anchored sequence leads to better performance. For LUMOS, the algorithm samples multiple truncated action sequences uniformly at random. Fig. 7 (right) shows that when these sequences are instead source-anchored, performance decreases compared to using randomly sampled sequences, which supports our choice. These results emphasize the critical role of selecting a good action sequence.

## 6 Related Work

**Transfer Learning.** Empirical methods for transfer learning in RL often lack provable guarantees or rely on application-specific assumptions that do not generalize. For instance, Sun et al. (2022) assume shared latent dynamics across observation spaces and that target observations can be transformed into the source domain, they do not learn this mapping explicitly and offer no theoretical guarantees on transfer performance. Similarly, Watahiki et al. (2024) assume a shared latent MDP structure between tasks but their method is not theoretically grounded. Other works, such as Agarwal et al. (2022), study transfer through reusing prior policies across design iterations, which is orthogonal to our setting. Methods like Chen et al. (2024); Yi et al. (2023) rely on very specific visual or object-level assumptions, which limits their general applicability. The notion of an *undo* map to reverse state space transformations in reinforcement learning was first introduced in Gupta et al. (2022), where the problem is approached through a distributional lens by aligning trajectories across the source and target tasks. However, the method does not scale beyond toy tabular environments.

**Representational Transfer in Low-Rank MDPs.** Several works have studied representational transfer under low-rank or linear MDP assumptions. For instance, Agarwal et al. (2023); Cheng et al. (2022) use reward-free exploration in the source task to learn a good representation for the target. Sam et al. (2024) extend this to a more general low-rank setting. Lu et al. (2021) learn a linear representation using least squares for multitask linear MDPs, and Ishfaq et al. (2024) consider the offline multitask case.

In contrast, our work approaches transfer RL through a more principled lens by making concrete assumptions about the structural relationship between the source and target tasks. We propose an algorithm that explicitly learns a mapping between their observation spaces by framing the problem as supervised learning, and therefore eliminates the need for RL in the target task. Under certain assumptions, this leads to provable improvements in sample complexity over learning from scratch. Moreover, our method is practical and demonstrates strong empirical performance. Essentially, our work takes a step toward more principled and broadly applicable approaches to transfer in RL.

## 7 Concluding Discussion

In this paper, we introduced a principled approach to transfer learning in reinforcement learning, where the source and target tasks are related by transformations of the state space. By explicitly learning an undo map, our method achieves significant gains in sample efficiency compared to learning target policies from scratch, as demonstrated across multiple challenging continuous control environments. Theoretically, we showed that when the undo map is linear and the source is linearly-$Q^\star$ realizable, our approach achieves strictly better sample complexity than learning from scratch. A current limitation of our analysis is the reliance on a *good* action sequence – one that satisfies specific coverage properties required to estimate the undo map accurately. A promising direction for future work is to design principled algorithms that can discover such sequences. Extending the theoretical results beyond the linear realizability assumption is another interesting direction.

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

# A   TABLE OF CONTENTS

In this section, we provide an outline of the contents provided in the paper's appendices.

- Appendix B contains the proof for Theorem 1.

- Appendix C describes the environments used in the experimental evaluation.

- Appendix D provides additional implementation details for experimental evaluation.

# B   PROOFS

In this section, we provide the proof of Theorem 1.

We first relate the performance difference between the optimal policy in the target MDP and the policy that acts greedily with respect to $\Phi_T \widehat{U}^T \theta_S^\star$ to the estimation error in $\widehat{U}$, where $\widehat{U}$ is the output of Algorithm 1:

**Lemma 1** (Performance Difference). $\|V_T^\star - V_T^{\hat{\pi}}\|_\infty \leq 2H\|\widehat{U} - U_\star\|_{\max}$.

Here, $\|\cdot\|_{\max}$ denotes the max norm. To bound the estimation error $\|\widehat{U} - U_\star\|_{\max}$, we use the following per-row guarantee:

**Lemma 2** (Estimation Error Bound). *Let $M[i]$ denote the $i$-th row of a matrix $M$. Fix $i \in [d_S]$. Then, for all $n \geq \frac{6}{H\lambda_{\min}}(\log \frac{3d_T}{\delta})$, with probability at least $1 - \delta$, the output $\widehat{U}$ of Algorithm 1 satisfies:*

$$\left\|\widehat{U}[i] - U_\star[i]\right\|_\Sigma \leq \sqrt{\frac{1}{nH}} \sqrt{d_T + 2\log\left(\frac{3}{\delta}\right) + 2\sqrt{d_T \log\left(\frac{3}{\delta}\right)}} + o\left(\sqrt{\frac{1}{nH}}\right).$$

Since $\Sigma$ is positive semidefinite (PSD), we can relate the $\Sigma$-norm to the $\ell_\infty$ norm via:

$$\|x\|_\infty \leq \frac{\|x\|_\Sigma}{\sqrt{\lambda_{\min}}}. \tag{10}$$

Applying a union bound over all $i \in [d_S]$, with per-row failure probability set to $\delta/d_S$, gives a high-probability bound on $\|U - U_\star\|_{\max}$. Substituting this into Lemma 1 concludes the proof.

**Proof of Lemma 1.** For any $s \in \mathcal{S}^T$,

$$\begin{aligned}
V_T^\star(s) - V_T^{\hat{\pi}}(s) &= Q_T^\star(s, \pi^\star(s)) - Q_T^\star(s, \hat{\pi}(s)) + Q_T^\star(s, \hat{\pi}(s)) - Q_T^{\hat{\pi}}(s, \hat{\pi}(s)) \\
&\leq Q_T^\star(s, \pi^\star(s)) - f(s, \pi^\star(s)) + f(s, \hat{\pi}(s)) - Q_T^\star(s, \hat{\pi}(s)) \\
&\quad + \mathbb{E}_{s' \sim \mathbb{P}^T(s, \hat{\pi}(s))}\left[V_T^\star(s') - V_T^{\hat{\pi}}(s')\right] \\
&\leq 2\|f - Q_T^\star\|_\infty + \mathbb{E}_{s' \sim \mathbb{P}^T(s, \hat{\pi}(s))}\left[V_T^\star(s') - V_T^{\hat{\pi}}(s')\right].
\end{aligned}$$

Unrolling this recursion for $H$ steps,

$$\|V_T^\star - V_T^{\hat{\pi}}\|_\infty \leq 2H\|f - Q_T^\star\|_\infty.$$

Setting $f = \Phi_T \widehat{U}^\top \theta_S^\star$ and $Q_T^\star = \Phi_T U_\star^\top \theta_S^\star$ completes the proof.

**Proof of Lemma 2.** Note that the noise in Algorithm 1 is 1-subgaussian. Therefore, the result follows from Theorem 1 and Remark 9 in (Hsu et al., 2014), where Condition 1 is satisfied with $\rho_0 \leq \frac{1}{\sqrt{\lambda_{\min} d_T}}$, Condition 2 is satisfied with $\sigma = 1$, and Condition 3 is satisfied with $b_0 = 0$.

## C  ENVIRONMENTS

**ROTATEDSWIMMER.** This task requires the agent to swim forward through a viscous fluid. We apply a $45°$ rotation to the $(x, y)$ plane to modify the frame in which the agent's motion is observed. This affects both the heading of the swimmer and its linear velocity vector. The observation is updated accordingly by adjusting the front tip's angle and rotating the $(x, y)$ velocity components.

**ROTATEDHALFCHEETAH.** This task requires the agent to run forward in a planar environment. The original environment operates in a 2D plane. We perform a $45°$ rotation on the $(x, z)$ components of the agent's torso position and velocity. The joint states and angular velocities remain unchanged.

**ROTATEDWALKER2D.** This task requires the agent to walk forward while maintaining balance. Similar to HalfCheetah, we rotate the observation of the agent's torso height and linear velocity in the $(x, z)$ plane by $45°$. This alters how progress and vertical motion are perceived, while the actuator and sensor states are kept intact.

**ROTATEDANT.** In this task, the agent is a quadraped and must navigate a 2D surface using four legs. We rotate the global $(x, y)$ velocity of the torso by $45°$. When present, external contact forces are also rotated in the same plane. The result is a consistent shift in perceived motion direction across the ant's high-dimensional observation space.

**Sensor Fusion.** In the fusion environments, instead of observing a $d_S$-dimensional observation vector, the agent receives a $10 \times d_S$-dimensional observation. More concretely, for each source observation element $i$, the target task provides 10 measurements of the form: $x_i^{(j)} = w_j \cdot x_i + c_j$, for $j \in [10]$, where the weights $w_j$ sum to 1 and the offsets $c_j$ sum to 0. This simulates a sensor fusion scenario in which the undo map $u_\star$ sums the 10 measurements corresponding to each original dimension to recover the source observation.

**POLARHALFCHEETAH.** In this task, we change the coordinate system of selected observation dimensions from Cartesian to polar coordinates. In particular, pairs of state variables such as the angular velocities of the back thigh and shin are transformed so that their joint $(x, y)$ representation becomes $(r, \theta)$, where $r = \sqrt{x^2 + y^2}$ and $\theta = \arctan 2(y, x)$. This alters how the agent perceives limb velocities.

**POLARANT.** In this task, the angular velocity components of the torso along the $x$ and $y$ axes are represented in polar form. This affects how rotational motion is encoded.

**PIXELCARTPOLE.** In this task, the agent receives pixel observations instead of low-dimensional sensor states. We extract a cropped $40 \times 60$ image centered around the cart, and the observation is the difference between two consecutive frames to capture dynamics. Since the cart's absolute position and velocity cannot be inferred from pixels alone, we mask those variables from the source task to ensure comparability. This setting, inspired by Sun et al. (2022), evaluates whether our method can transfer across a modality shift from sensor states to pixels.

## D  IMPLEMENTATION DETAILS

**Rotation and Fusion Tasks.** To estimate the expected state feature sums in the target MDP $\mathcal{M}T$, we truncate rollouts to $d_T$ steps instead of executing full episodes. This produces a sufficient number of equations to identify the undo map. We collect approximately $5,000$ rollouts of length $d_T$, where $d_T$ is the dimensionality of the target observation space. A comparable number of samples is used to estimate the source statistics. The action sequence $a_{0:H-1}$ is obtained from the source policy $\pi_S$: we execute a rollout of $\pi_S$ in the source MDP $\mathcal{M}S$, record the actions taken, and reuse this sequence as $a_{0:H-1}$.

**Polar Tasks.** For these tasks, we generate 500 randomly sampled action sequences of length $d_T$. The target statistics $\bar{\psi}_{h,\cdot}^T(\cdot)$ are estimated using 10 samples, while the source statistics $\bar{\psi}_h^S(\cdot)$ are estimated using 10 samples for POLARHALFCHEETAH and 3 samples for POLARANT.

**Sensor-to-Pixel Task.** Here, we use 500 source-anchored action sequences of length 5. The target statistics $\bar{\psi}_{h,\cdot}^{T}(\cdot)$ are estimated with 500 samples, while the source statistics $\bar{\psi}_{h}^{S}(\cdot)$ are estimated with 20 samples.

For LUMOS-LIN, the undo map is computed using the closed-form least-squares solution. For LU-MOS, we parameterize the undo map with a single hidden-layer neural network, trained for 10,000 epochs with learning rate 0.001 and weight decay 0.0001. The hidden layer size is set to 128 for POLARHALFCHEETAH, 256 for POLARANT, and 16 for PIXELCARTPOLE.

