# OpenReview forum: "Learning to Undo: Transfer Reinforcement Learning under State Space Transformations"
_ICLR.cc/2026/Conference — ICLR 2026 Conference Withdrawn Submission_

### Official Review · Reviewer_enW4 · 2025-10-23

**Soundness:** 1
**Presentation:** 3
**Contribution:** 2
**Rating:** 2
**Confidence:** 3

**Summary:**

The paper proposes a transfer learning method, capable of recovering state-space transformations between target $\mathcal{M}_t$ and source tasks $\mathcal{M}_s$. Two settings for the algorithm LUMOS are proposed: one with theoretical guarantees, another with empirical evidence.

**Strengths:**

The paper is well written, and present a novel although narrow perspective for transfer learning in Reinforcement Learning.

**Weaknesses:**

Theoretical results rely on a strong assumption given by Eq. 3. Practically, it does not seem feasible to tell beforehand if the state-space and dynamics of $\mathcal{M}_t$ are a transformation (linear or not) of the state-space and dynamics of  $\mathcal{M}_s$, which heavily limits the applicability of the method. Empirical results do not explore this weakness. As such, the paper’s contribution seems reduced to finding a map between two MDPs' dynamics, presuming an oracle (otherwise, fail). Thus, the last sentence before the concluding sentence appears misleading.

In the Related Works, authors have pointed out limitations of related works regarding theoretical guarantees, while the proposed method itself covers the narrow setting of linear state-space mappings. The arguments are not convincing due to the lack of baselines in the empirical evaluation, other than learning the task from scratch.

The manuscript is overall well written, but some parts are a bit confusing, such as the open loop definition in section 3.1, and short sentences like “We assume that $d \le H$”.

**Questions:**

**Q1.** In section 2 - transfer learning, and algorithm 2, during transfer, interacting with source tasks does not incur sample complexity. Would it be feasible to compare other metrics than sample complexity to assess the cost related to transferring (and interacting with $\mathcal{M}_s$)?

**Q2.** In Related Works, in the very last sentence, the author claims that the work  _“takes a step toward more principled and broadly applicable approach...”_. How does it hold with the strong assumption of linear transformation state-spaces?

---

### Official Review · Reviewer_7ZNp · 2025-10-25

**Soundness:** 3
**Presentation:** 3
**Contribution:** 2
**Rating:** 4
**Confidence:** 3

**Summary:**

This paper studies transfer learning in reinforcement learning, focusing on when and how knowledge can be transferred between Markov Decision Processes (MDPs) to improve learning efficiency. The authors consider the setting where there exists an undo map between a source MDP and a target MDP, such that applying this map to the target’s state space exactly recovers the source.

The main contributions of the paper are:
1. Algorithm for learning the undo map: The paper proposes a method that learns this map via regression on state feature statistics collected from both source and target MDPs. Once learned, the map enables zero-shot transfer of the source policy to the target.

2. Theoretical analysis: When the undo map is linear and the source MDP is linearly realizable, the proposed approach achieves provably better sample complexity than learning the target MDP from scratch.

3.  Empirical validation: Experiments on challenging continuous control tasks demonstrate that the approach achieves significantly higher sample efficiency than baseline methods.

**Strengths:**

1. Originality: Introduces the novel concept of an undo map to reverse state-space transformations, enabling zero-shot policy transfer.

2. Quality: Methodologically rigorous with theoretical guarantees in the linear setting. Algorithms are clearly defined, and experiments on diverse continuous control tasks show substantial improvements in sample efficiency.

3. Clarity: Well-structured and readable. Algorithms, definitions, and assumptions are clearly presented, with helpful figures illustrating key ideas.

Significance: learning high-quality policies from scratch typically requires millions
of environment interactions.

**Weaknesses:**

1. Limited Baselines: Experiments mainly compare against learning from scratch. To better demonstrate the practical advantage of proposed method, the paper should include comparisons with existing transfer learning RL methods, such as Chen et al. (2024) or other policy-transfer approaches. This would strengthen claims of empirical superiority.

2. Theoretical Scope: The theoretical analysis is restricted to linear undo maps and linearly-Q⋆ realizable MDPs. While LUMOS extends to non-linear transformations empirically, the lack of theoretical guarantees in this general setting limits the rigor of the contribution. Extending bounds or providing analysis for non-linear settings would significantly enhance impact.

3. Action Sequence Dependence: Both LUMOS-LIN and LUMOS require carefully chosen action sequences for effective learning of the undo map. The performance is sensitive to these sequences, but no principled method for discovering or optimizing these sequences is provided. This work could include a more systematic approach for generating action sequences.


[1] Chen, E., Chen, X., & Jing, W. (2024). Data-Driven Knowledge Transfer in Batch $ Q^* $ Learning. arXiv preprint arXiv:2404.15209..

**Questions:**

Is the sample complexity upper bound of Algorithm 2 tight?

---

### Official Review · Reviewer_J82C · 2025-10-31

**Soundness:** 3
**Presentation:** 2
**Contribution:** 2
**Rating:** 2
**Confidence:** 4

**Summary:**

This paper proposes LUMOS, a method for transfer RL, when source and target tasks differ only by a transformation of the state. It learns an undo map that converts target observations into the source feature space so a source policy can be reused. The authors provide theory for the linear case and extend the method to nonlinear settings. Experiments on continuous-control tasks show it is more sample-efficient than training from scratch.

**Strengths:**

The paper is technically sound and provides a clear theoretical analysis under the linear-Q-realizable setting, with consistent assumptions and proofs. Experiments on multiple tasks validate the effectiveness of their method. The overall structure is logical and easy to follow, and the figures and algorithms are well presented.

**Weaknesses:**

1. The paper builds on the "undo map" formulation. While this setting is well-defined and analytically tractable, and linearity assumption in theory is acceptable, it represents a relatively narrow class of transfer scenarios, where tasks dynamics are identical and only differ by an invertible transformation. This strong assumption limits its contribution to the current transfer RL research, where most problems are more complicated and no such deterministic "undo map" exists. The paper motivates the undo map using a few clean geometric and sensor transformation examples, but these settings may not capture the challenge in most transfer situations.
2. The paper repeats several key ideas (such as their contributions) too many times across multiple sections, with only minor variations. Streamlining these explanations would make the presentation clearer and more concise.
3. The paper does not include any ablation studies testing how key design choices affects the transfer results. For instance, examining how the choice of action sequence, target sample budget, or source policy quality affects the accuracy of the learned undo map.
4. The paper does not provide open-source code and includes limited experimental details, which reduces reproducibility and transparency.

**Questions:**

1. How might the proposed method be extended or adapted to handle more general cases, where an exact undo map does not exist?
2. In the related work section, the paper claim that their method is practical and demonstrates strong empirical performance. Could you clarify how this comparison is fair, given that many related methods tackle more general transfer settings than the invertible-transformation case studied here?
3. The proposed method relies on accurately matching feature statistics between the source and target tasks. How sensitive the approach is to potential noises in the target observations?
4. In Equation 1, $\psi_h (a_{0:h-1})$ is defined as the sum of state features, but in equation (5), $\psi_h (a_{0:h-1})$ is defined as the average of state-action features. This inconsistence is confusing.

---

### Official Review · Reviewer_zNyp · 2025-11-01

**Soundness:** 1
**Presentation:** 1
**Contribution:** 1
**Rating:** 2
**Confidence:** 4

**Summary:**

This paper proposes a method for transfer learning in RL that learns an "undo map" to transform target state spaces back to source state spaces via regression on state feature statistics. The transfer learning is studied only in the context where the observation space has changed and not the dynamics or rewards. Experiments are conducted on continuous control problems.

**Strengths:**

* Transfer learning is an important problem in RL.
* The proposed method is interesting and has potential.

**Weaknesses:**

* I found several overclaims by the authors and occasionally misleading comparisons:

1. The setting studied here represents a very narrow and limited form of transfer learning. The authors only consider changes in the observation space (e.g., a change in viewpoint), whereas transfer learning in the broader literature often involves changes in rewards, dynamics, or both. Even within this limited setting, the assumed transformation must follow a specific linear structure, and the experiments are deliberately designed to satisfy this assumption (e.g., the sensor fusion experiments). The RGB-to-grayscale transformation mentioned in the introduction is missing from the experiments, and no experiments on naturally occurring transformations are provided. As a result, statements such as the following are clear overclaims:
> (L463) Our method achieves significant gains in sample efficiency compared to
learning target policies from scratch, as demonstrated across multiple challenging continuous control environments.

2. The authors repeatedly describe their method as a zero-shot transfer learning approach, yet it explicitly requires thousands of samples from the target task to learn the undo map. In the literature, zero-shot transfer refers to settings where no samples from the target task are available.

3. The authors claim novelty in proposing a new method for transfer learning (L80), yet the idea of using undo maps for transfer was already introduced by Gupta et al. (2020). The only apparent difference here is that regression on feature statistics is used instead of trajectory matching.

4. Empirical overclaims include statements such as the following. However, this comparison is made against learning from scratch, which is not a transfer learning algorithm and has no access to the source task or policy.
> (L411) LUMOS-LIN is 143× more sample-efficient.

5. The paper compares the sample complexity of the proposed method against the exponential lower bounds of Weisz et al. (2021), even though that work concerns worst-case MDPs specifically constructed to be difficult. For linear MDPs—the focus of this paper’s theoretical section—sample complexity bounds are already significantly better (see: Jin et al., “Provably efficient reinforcement learning with linear function approximation,” COLT 2020). Thus, the authors’ claim of “strictly better sample efficiency” is misleading; the improvement is marginal and applies only to a very narrow problem class, which should be clearly stated.

6. The practical algorithm deviates considerably from the theoretical setting—for instance, it uses random action sequences instead of source rollouts. The paper provides no justification for why matching statistics should work in non-linear settings. Therefore, the following statement is also an overclaim:
> (L67) Our algorithmic design is grounded in the setting where the undo map is linear and the source…



* The paper has a limited experimental setup:
1. As noted above, the environments are often artificially constructed to satisfy the assumptions of the method.

2. The method is not compared against any prior algorithms from the transfer learning, meta-RL, or domain adaptation literature. The only baseline used is training from scratch, which raises major concerns about the validity of the claimed performance gains. At a minimum, the paper should include one baseline from each of these families to contextualize the proposed approach. Given that the transfer tasks involve only changes in the observation space, even simple methods such domain adaptation methods or simple data augmentation could serve as competitive baselines.

3. The experiments use only three random seeds, which is insufficient for RL benchmarks. Moreover, it is unclear what the shaded regions in the plots represent.

* Methodological issues:
1. Figure 7 shows that the method can completely fail when using random action sequences for certain tasks. This critical dependency is understated and not properly discussed.

2. The paper assumes an identifiable undo map from statistics, but the notion of identifiability is neither defined nor analyzed.

* The paper is not well-placed within the extensive literature on transfer in RL, omitting several key references, including but not limited to:

* Touati, Ahmed, Jérémy Rapin, and Yann Ollivier. "Does Zero-Shot Reinforcement Learning Exist?." The Eleventh International Conference on Learning Representations.
* Touati, Ahmed, and Yann Ollivier. "Learning one representation to optimize all rewards." Advances in Neural Information Processing Systems 34 (2021): 13-23.
* Rezaei-Shoshtari, Sahand, et al. "Hypernetworks for zero-shot transfer in reinforcement learning." Proceedings of the AAAI Conference on Artificial Intelligence. Vol. 37. No. 8. 2023.
* Beck, Jacob, et al. "Hypernetworks in meta-reinforcement learning." Conference on Robot Learning. PMLR, 2023.
* Rakelly, Kate, et al. "Efficient off-policy meta-reinforcement learning via probabilistic context variables." International conference on machine learning. PMLR, 2019.
* Arndt, Karol, et al. "Meta reinforcement learning for sim-to-real domain adaptation." 2020 IEEE international conference on robotics and automation (ICRA). IEEE, 2020.
* Ju, Hao, et al. "Transferring policy of deep reinforcement learning from simulation to reality for robotics." Nature Machine Intelligence 4.12 (2022): 1077-1087.
* Ingebrand, Tyler, Amy Zhang, and Ufuk Topcu. "Zero-shot reinforcement learning via function encoders." Proceedings of the 41st International Conference on Machine Learning. 2024.

**Questions:**

1. What happens when the source policy is sub-optimal?

---

### Note · Authors · 2025-12-03

**Comment:**

We thank the reviewers for carefully reviewing our paper! We greatly appreciate the feedback. Please see below our common response to the comments.

---

**Problem Setup**

Our goal in this work is to study a transfer learning setting in RL where one can obtain provably better sample complexity than learning from scratch. While several existing transfer RL algorithms work in more general settings, they do not come with any sample complexity guarantees. Therefore, we focus on a practical setting: state space transformations, which capture several relevant scenarios (e.g., sensor space to pixel space representations in control tasks).

---

**Method**

Given this setup, we propose a concrete transfer RL algorithm based on matching state-feature statistics. When the transformation is linear and the target MDP is linearly-$Q^\star$ realizable, this method can be shown to be provably more sample efficient than learning from scratch. Empirically, it also performs well beyond the assumptions used in the analysis.

---

**Empirical Evaluation**

In the experiments, we compare this method against learning from scratch because the goal of the paper is not to propose a new SOTA empirical transfer RL algorithm, but to demonstrate that transfer learning can be more sample efficient in a principled setting, both theoretically (under assumptions) and empirically (in continuous-control tasks).

---

We will use the feedback to improve the work. Thank you again for the reviews.

**Withdrawal Confirmation:**

I have read and agree with the venue's withdrawal policy on behalf of myself and my co-authors.